# Interpreting the Benefit and Risk Data in Between-Drug Comparisons: Illustration of the Challenges Using the Example of Mefenamic Acid versus Ibuprofen

**DOI:** 10.3390/pharmaceutics14102240

**Published:** 2022-10-20

**Authors:** André Farkouh, Margit Hemetsberger, Christian R. Noe, Christoph Baumgärtel

**Affiliations:** 1Department of Pharmaceutical Sciences, University of Vienna, 1090 Vienna, Austria; 2Hemetsberger Medical Services, 1070 Vienna, Austria; 3Department of Medicinal Chemistry, University of Vienna, 1090 Vienna, Austria; 4AGES Austrian Medicines and Medical Devices Agency, Austrian Federal Office for Safety in Health Care, 1200 Vienna, Austria

**Keywords:** ibuprofen, mefenamic acid, non-steroidal anti-inflammatory drugs, pain medication, between-drug comparison

## Abstract

Evidence-based pain therapy should rely on precisely defined and personalized criteria. This includes balancing the benefits and risks not only of single drugs but often requires complex between-drug comparisons. Non-steroidal anti-inflammatory drugs (NSAIDs) have been available for several decades and their use is described in an abundance of guidelines. Most of these guidelines recommend that ‘the selection of a particular NSAID should be based on the benefit-risk balance for each patient’. However, head-to-head studies are often lacking or of poor quality, reflecting the lower standards for clinical research and regulatory approval at the time. The inconsistency of approved indications between countries due to national applications adds to the complexity. Finally, a fading research interest once drugs become generic points to a general deficit in the post-marketing evaluation of medicines. Far from claiming completeness, this narrative review aimed to illustrate the challenges that physicians encounter when trying to balance benefits and risks in a situation of incomplete and inconsistent data on longstanding treatment concepts. Ibuprofen and mefenamic acid, the most frequently sold NSAIDs in Austria, serve as examples. The illustrated principles are, however, not specific to these drugs and are generalizable to any comparison of older drugs in daily clinical practice.

## 1. Introduction

Pain is not a simple, precisely defined neuronal sensation but rather a complex series of pathophysiological, emotional, and behavioral processes. An Austrian survey among patients suffering from pain reflects this complexity when reporting that around 40 to 65 percent of respondents were dissatisfied with their treatment and frustrated by the care received [1,2]. A comprehensive assessment of an individual’s needs and a tailored, multimodal, and interdisciplinary treatment strategy in accordance with international standards are thus warranted [3,4,5]. Especially in the treatment of acute pain, rapid therapy within the limits of the drug’s pharmacokinetic properties is important to avoid the occurrence of manifest changes in the central nervous system and thus pain chronification [6,7]. Chronic pain, however, is best treated at a fixed administration schedule allowing for a stable effect throughout the day [6,7].

Non-steroidal anti-inflammatory drugs (NSAIDs) represent a large class of pain killers, of which most representatives have been available for several decades. However, recent research, especially high-quality head-to-head trials are often lacking, reflecting the lower standards for clinical research and regulatory approval at the time these drugs came to market. This complexity of pain management at all stages underlines the importance of comprehensive and evidence-based consultation at the patients’ point of contact, i.e., the doctor’s office or the pharmacy.

To reduce complexity while providing up-to-date clinical evidence, well-maintained guidelines are available to physicians and pharmacists [3,8,9,10]. Most guidelines recognize this unsatisfactory situation and shift the responsibility for individual assessment of the benefit-risk balance to the consultant. A typical statement is the following [10]: ‘Overall, most evidence showed no significant efficacy differences among the different non-selective NSAIDs. However, the quality of evidence for the majority of indications was limited by small and underpowered studies with imprecise estimates of effect and the absence of several randomized, controlled trials (RCTs) for the same non-selective NSAID comparison. Additionally, RCTs for every possible NSAID comparison are lacking. Guidelines, across indications, do not specify a preference for one non-specific NSAID over another.’ Sometimes this is followed by a disclaimer, such as the following [3]: ‘Practitioners should choose medication within their appropriate prescribing rights and within their scope of professional practice and accept clinical/legal responsibility for their prescribing decisions.’ The present narrative literature review aimed to illustrate the challenges that physicians encounter when trying to balance benefits and risks in a situation of incomplete and inconsistent between-drug comparisons of longstanding treatment concepts. Ibuprofen and mefenamic acid in their locally approved Austrian indications have been chosen as examples. This review is not trying to provide a complete summary of the literature available to date but rather to illustrate a principle, which is generalizable to any comparison of older drugs in daily clinical practice.

## 2. Pain Management in the Course of Time

NSAIDs are the largest and most commonly prescribed type of analgesic. Many compounds are available prescription-free through pharmacies or over-the-counter (OTC) for self-medication. As a general consideration it should be noted that all substances that have obtained national or regional regulatory approval were considered effective and safe in their respective indication by the regulatory authorities [11]. In Austria, the NSAIDs acetylsalicylic acid, paracetamol and ibuprofen are OTC drugs, while mefenamic acid requires a prescription [12]. In other countries, however, OTC preparations for mefenamic acid are available. Reference is made to local regulatory agencies for the respective prescribing information.

According to a report by the Austrian Federal Environment Agency [13], the total consumption of analgesics, anti-inflammatory drugs and anti-rheumatic agents in Austria in 2014 was 244,854 kg. Compared to 1997, this represents an increase of 50%. There was also a shift in the individual compounds most frequently used: while in 1997 acetylsalicylic acid, paracetamol and mefenamic acid were among the three most commonly used compounds in Austria, acetylsalicylic acid was increasingly replaced by ibuprofen by 2014 [13,14]. Among NSAIDs, this leaves ibuprofen and mefenamic acid as preferred choices in Austria, often for similar indications. Lacking comprehensive head-to-head trials, the choice of compound is also often guided by individual experience and preference of the physician, the pharmacist and even the patient. Even the comprehensive compendium “Martindale—the Complete Drug Reference” points out that the medical prescription is often based only on doctors’ experience [15].

## 3. History of NSAIDs and Its Impact in Data Quality Today

NSAIDs represent one of the oldest classes of medicines. They are typical representatives of the era of chemistry-driven drug research, which was dominant until the middle of the 20th century and was eventually replaced by targeted (pharmacology-driven) drug design. Acetylsalicylic acid is a typical example of chemistry-driven drug design. It was initially developed as a prodrug of salicylic acid, designed with the aim to avoid the adverse effects of its parent compound. It was only much later found to be an irreversible enzyme blocker perfectly suited as antithrombotic drug. Mefenamic acid received its first market approval in 1962 in the United States [16]. Ibuprofen was first approved in the United Kingdom in 1969 [17]. At the time of market entry, the basic mechanisms of action were not yet known. The discovery of the role of the arachidonic acid cascade and of the prostaglandins by Bergström, Samuelson, and Vane was rewarded by the Nobel Prize in Medicine in 1982. Prostaglandin synthase, more frequently referred to as cyclooxygenase (COX), is responsible for the transformation of arachidonic acid to the prostaglandins, prostacyclin and thromboxane. It occurs in two isoforms: COX-1 and COX-2 [18], which were only discovered in 1971 [19] and 1990/91 [20,21], respectively. In the meantime, extensive pharmacological work on the arachidonic acid cascade has revealed the high complexity of this system, which increasingly will be recognized to be the core of innate immunity.

This iterative procedure stands in considerable contrast to the common drug discovery process today, where a comprehensive package of data on a drug is generated and submitted to inform the regulatory approval process. Marketing approval follows a systematic assessment of all available information on a drug. However, regulatory requirements have evolved in parallel with the scientific progress and also as a consequence of dramatic failures, such as the discovery of the malformations caused by thalidomide in the 1950s and 1960s [22]. Even now, marketing approval is often granted on a preliminary basis with further requests for post-authorization safety studies (PASS), because rare side effects, interactions or risks that only affect certain patient groups only become apparent when a large number of individuals have received the new drug. Available data on old drugs are therefore often incomplete and of low quality as per current standards. Furthermore, with each new chemical entity (NCE) introduced to the market becoming a generic drug after some time (10 to 15 years), the majority of medicines in use consists of products, which have never been subject to a systematic comparative post-marketing assessment guided by drug authorities according to current standards.

## 4. Mechanism of Action of NSAIDs

All NSAIDs act by inhibiting COX-1 and/or COX-2 [23]. COX-1 is the constitutive COX isoform and is mainly responsible for the production of prostaglandins. This role is often referred to as the “housekeeping” role. Acetylsalicylic acid irreversibly binds to COX-1 [19]. COX-2 is the inducible isoform that is stimulated as a result of tissue damage or inflammation. Pharmacologically inhibiting COX-2 produces an anti-inflammatory effect. The main symptoms of inflammation are redness, heat, swelling, and pain. Prostaglandin E_2_ (PGE2), a prostaglandin formed as a result of COX stimulation, is involved in all of these key symptoms. PGE2 is formed via both COX-1 and COX-2. Pain is caused by the effects of PGE2 on the peripheral nervous system, spinal cord and brain [24]. Via this mechanism, the synthesis of PGE2 is also inhibited and thus inflammation and pain is reduced [24].

## 5. Ibuprofen and Mefenamic Acid

Following this brief history of the development of NSAIDs in the context of the evolution of regulatory procedures over time, the implication of the resulting lack of good quality data on the assessment of benefits and risks of frequently prescribed drugs today is illustrated using the example of the arylpropionic acid derivative ibuprofen as compared to the anthranilic acid derivative mefenamic acid. A pattern of quality issues and missing and/or contradictory data will emerge.

### 5.1. Pharmacokinetics

Administered orally, mefenamic acid is well absorbed through the gastrointestinal tract [15]. The peak plasma concentrations (C_max_) and plasma half-life (t_1/2_) are reached after 2–4 h. Over 90% is bound to protein and distribution into breast milk has been observed. Mefenamic acid is metabolized via CYP2C9, but about 50% is excreted in the urine unchanged or as metabolite conjugates; fecal excretion is about 20%.

Ibuprofen is commonly administered orally, but there are also rectal suppositories, topical gels, and intravenous formulations. After oral administration, ibuprofen is rapidly and completely absorbed, the percutaneous absorption is low at a level of approximately 5% [15]. Therapeutic concentrations are reached already after 30 min. C_max_ is reached after 1–2 h, and plasma t_1/2_ is approximately 1.8 to 2.44 h [10,25]. Ibuprofen is extensively bound to plasma protein (>90%). The volume of distribution (V_d_) depends on age and body temperature and ranges from 0.12 L/kg to 2 L/kg. Ibuprofen is excreted primarily via the urine (45 to 79%) within 24 h, mainly as metabolites and as free (1%) or conjugated (14%) ibuprofen, with small amounts of biliary excretion [15]. Table 1 provides an overview of important pharmacokinetic parameters.

#### 5.1.1. Different Formulations of Ibuprofen

Distinct galenic formulations of ibuprofen with different pharmacological properties [26,27] and different approved indications [11] exist. Different effects on the gastrointestinal mucosa have been reported. For example, the sodium salt showed higher gastric mucosal irritation in rats than the parent compound [28], probably due to higher water solubility and increased mucosal absorption. The lysin salt showed higher absorption in humans, with lower gastric irritation than acetylsalicylic acid [29], but higher gastric irritation than placebo [30]. However, a recent study showed that the onset of action of ibuprofen lysinate was not significantly faster than ibuprofen acid [31].

#### 5.1.2. Ibuprofen and Dexibuprofen

Synthetic drugs—in contrast to natural products—were traditionally available as racemates. However, according to the lock-key-principle, only one enantiomer is expected to exhibit pharmacological activity, bringing the topic of eutomer and distomer (racemic burden) into the focus. In the last quarter of the 20th century, progress in chemical methods allowed the synthesis of pure enantiomers. This eventually lead to the development of the pharmacologically active S(+)-enantiomer, dexibuprofen, which is now marketed as a drug of its own.

#### 5.1.3. Comparative Pharmacology of S(+)-Ibuprofen and R(−)-Ibuprofen

Ibuprofen is a racemate of equal parts of the S(+)- and R(−)-enantiomer. Adults slowly convert about 60% of R(−)-ibuprofen to S(+)-ibuprofen. This transformation occurs to a lesser extent in children [32]. S(+)-ibuprofen is metabolized by hepatic oxidation primarily by CYP2C9 to inactive metabolites. This step can be followed by phase II eliminations, in which the oxidative metabolites are conjugated to glucuronide before excretion.

The two enantiomers of ibuprofen also differ in their pharmacological properties, leading some authors to classify them as different drugs [33]. There is evidence for systemic—mainly hepatic—unidirectional inversion of the R(−)-enantiomer into the S(+)-enantiomer in humans [34], which was not observed in the other direction [35], thus tilting the balance between the R(−)- and S(+)-enantiomers. Pre-systemic conversion was not observed in vivo [36]. The systemic nature of enatioconversion may explain differences in effectiveness and the side-effect profile of racemic ibuprofen compared to dexibuprofen. For instance, the R(−)-enantiomer seems to be involved in lipid metabolism pathways, while the S(+)-enantiomer is not. On the other hand, only the S(+)-enantiomer is capable of inhibiting COX at clinically relevant concentrations to induce pain relief through inhibition of prostaglandin production [24,37]. However, COX inhibition is thought to be the main cause of NSAID-induced gastric mucosal injury [37]. It is hypothesized that both enantiomers compete to bind the active sites of the COX isoenzymes in the stomach and intestinal mucosa cells [38,39]. R(−)-ibuprofen does not bind to COX but is capable of masking the COX binding sites to inhibit binding of the S(+)-enantiomer, which results in a lower rate of gastric adverse effects and reduces ulcerogenic activity and bleeding [39]. Taken together, these properties explain the clinical benefits of dexibuprofen over racemic ibuprofen, such as its greater clinical efficacy and lower variability in therapeutic effects [33,40], while the racemic form may have a lower gastrointestinal toxicity [39].

A prospective clinical trial under the supervision of authorities comparing the effects of the double dose of ibuprofen with a single dose of dexibuprofen might provide valuable additional information not only for ibuprofen, but also for racemates in general. Such studies have relatively complex study designs as they must take into account the inversion phenomenon when the racemic drug is administered [36,41,42].

### 5.2. Therapeutic Indications

This review is limited to oral ibuprofen or mefenamic acid; parenteral, topical, rectal and other forms of administration, as well as combination preparations are not considered here. Indications approved in Austria were considered and package inserts were retrieved from the Austrian Registry of Proprietary Medicinal Products of the Federal Office for Safety in Health Care—Austrian Medicines and Medical Devices Agency [11]. Both products are available in generic and/or proprietary forms in many countries worldwide and reference is made to local labels for the respectively approved therapeutic indications of each product and brand.

Ibuprofen is approved in Austria for a wide range of pain conditions and inflammatory diseases, whereby the labels of the individual brands and formulations differ widely [11]. It is used, for example, in pain conditions (e.g., back, tooth, muscle, joint and nerve pain, menstrual cramps, migraine), in pain in conjunction with colds and flu infections, in acute and chronic arthritis (rheumatoid arthritis), for arthrosis, inflammatory rheumatic diseases (ankylosing spondylitis, soft tissue rheumatism), as well as for painful swelling and inflammation.

Mefenamic acid is authorized and used in Austria for the symptomatic treatment of mild to moderate acute and chronic pain in rheumatic diseases, muscle pain, pain in the spine (e.g., intervertebral disc pain), pain, swelling and inflammation after injury or surgery, as well as pain in primary dysmenorrhea. For mefenamic acid, the indications approved in Austria do not differ between available brands [11].

Both ibuprofen and mefenamic acid are used in both adults and children. The administration of NSAIDs is consistently recommended for short-term administration at the lowest possible dosage, as the risk of serious side effects increases with the duration of administration and the dose—this applies to both cardiovascular risk and the risk of gastrointestinal side effects [15,43,44,45]. In the case of OTC preparations, a physician should be consulted when the treatment duration exceeds 10 days [8]. However, there is some evidence that e.g., with gastrointestinal problems, the highest risk occurs at the beginning of treatment [46].

The class of NSAIDs differs from other analgesics mainly by their anti-inflammatory properties. The inhibition of prostaglandin synthesis and thus the anti-inflammatory effect usually occurs quickly. In some diseases, however, it can take days to weeks for the inflammatory condition to improve substantially [15]. Contrary to popular belief, however, the antiphlogistic effect does not appear to be equally pronounced across all NSAIDs.

### 5.3. Safety Profiles

#### 5.3.1. Gastrointestinal and Cardiovascular Side Effects

NSAIDs are commonly known to exhibit gastrointestinal side effects (nausea, vomiting, gastrointestinal bleeding, etc.), as well as cardiovascular (hypertension, edema, heart attack, stroke, etc.), renal (acute kidney failure, hyperkalemia, fluid retention, etc.) and hepatic side effects (increased aminotransferase levels, hepatitis, liver failure), and they can cause allergic reactions (anaphylaxis).

Depending on their COX-1/COX-2 selectivity, the use of NSAIDs creates an imbalance between COX-1-mediated and COX-2-mediated effects. Prostaglandins, which are formed by COX-1 enzymes, serve to protect the gastric mucosa, vascular homeostasis, promote platelet aggregation and control kidney function. It is therefore generally accepted that inhibiting COX-1 inhibits the formation of blood clots, but plays an important role in the development of gastrointestinal and renal adverse effects [10,47]. The anti-inflammatory and analgesic effect of NSAIDs is COX-2 dependent, since COX-2 is the inducible COX isoform. In general, however, COX-2 inhibition is associated with higher cardiovascular toxicity and susceptibility to thrombosis [8,10]. COX-2 is also involved in the healing of existing ulcerations [48]. However, recent literature indicates that the degree of COX-2 selectivity does not appear to be the only determinant of cardiotoxicity [49]. Cardiotoxicity has been associated with differences in physiochemical properties between different NSAIDs [50,51,52]. An increase in susceptibility of cardiomyocyte membranes to oxidative damage has also been hypothesized [53], as has been an increase in a toxic metabolite of arachidonic acid [54]. More research is needed to better understand these physiochemical properties of different NSAIDs and to finally allow adequate clinical conclusions.

This also applies to the degree of COX-1 selectivity and gastrointestinal risk—at least for non-selective, traditional NSAIDs [45]. Especially with regard to the gastrointestinal risk, there is evidence that NSAIDs, which strongly inhibit both COX isoforms in therapeutic concentrations, have the highest gastrointestinal toxicity [48,55]. Often the duration and dose of therapy, the intra-individual variability of plasma levels, as well as existing risk factors play a greater role [10,45,49], which can be a problem especially in older patients, who often take NSAIDs as long-term medication [46,56].

Both substances, mefenamic acid and ibuprofen, pertain to the class of traditional COX inhibitors. They are characterized in vitro by a fast, competitive and reversible binding of COX-1 and COX-2 [57]. Ibuprofen is considered a non-selective COX inhibitor with medium COX-2 selectivity, while mefenamic acid, although a non-selective COX inhibitor, has a preference towards COX-2 [10,58]. As explained above, it is not permissible to conduct a benefit–risk assessment solely on the basis of the degree of COX selectivity of an NSAID.

For ibuprofen, a lower risk of gastrointestinal side effects is often assumed. However, this only seems to be the case at low doses, up to 1200 mg daily [59]. Numerous studies and reviews show that at higher doses, above 1600 mg daily, ibuprofen has the same incidence of gastroduodenal side effects as, for example, diclofenac or naproxen [60,61,62,63,64].

Both ibuprofen and mefenamic acid list cardiovascular events and gastrointestinal side effects in the warnings in their package leaflets [11]. The FDA generally warns of an increased cardiovascular risk when using NSAIDs [43]. A study by the EU Pharmacovigilance Risk Assessment Committee (PRAC) showed a dose dependence of cardiotoxicity for ibuprofen. For example, recommended OTC doses up to 1200 mg/day are not associated with any cardiovascular risk, but doses greater than 2400 mg per day show a slightly increased risk of cardiovascular events such as heart attack and stroke and should generally be avoided in patients with severe cardiovascular disease or in patients who had previously had a heart attack or stroke [44]. These recommendations for ibuprofen also apply to dexibuprofen. A high dose of dexibuprofen is considered to be a dose of 1200 mg or greater per day [44]. In view of the pharmacological activities discussed above, this means that the acceptable pharmacologically active dose of dexibuprofen is set at a level that is half of that of ibuprofen. In principle, a comparable safety profile, including contraindications and restrictions on use, of ibuprofen and dexibuprofen can be assumed [65]. Various publications postulate a comparable efficacy of dexibuprofen compared to racemic ibuprofen in the treatment of pain conditions of varying genesis as well as fever in children [65,66,67]. In its assessment, the PRAC found that while no specific data on the cardiovascular risk of dexibuprofen are available, a similar cardiovascular risk as with a high dose of ibuprofen can be expected when dexibuprofen is used in equipotent doses [67].

Importantly, NSAIDs in general should be used with caution in older adults with heart failure who are asymptomatic and avoided in those who are symptomatic [68].

#### 5.3.2. Neurotoxic and Psychiatric Effects

There have been reports of certain protective but also adverse neurological and psychiatric effects of NSAIDs. Under certain circumstances, most cell types in the central nervous system (CNS), including neurons and glia cells, have the capacity to express both COX-1 and COX-2 [69]. Randomized, controlled trials (RCTs), however, are rare and the available evidence therefore often comes from patient- or health care practitioner-reported adverse effects of overdose [70] or from case reports [71]. It is therefore difficult to establish causal relationships and discern the drug effect from the disease background and other confounding factors and a comparative assessment of the CNS toxicity of individual NSAIDs based on anecdotal evidence alone is not possible. Many CNS toxicities appear to be related to the decreased cerebral prostaglandin and thromboxane synthesis [69].

NSAIDs, including ibuprofen and naproxen, have been associated with cases of drug-induced aseptic meningitis [72,73,74,75] and there seems to be an allergic basis to this effect [76]. As will be discussed in more detail below, aspirin has a well-established anti-platelet effect but interference with this effect through NSAIDs, such as ibuprofen, has been documented [77,78]. The risk of stroke was, however, not increased for ibuprofen in a retrospective study of a cohort of 336,906 persons and 4354 stroke hospitalizations [79]. There is also a stroke model which showed a neuroprotective effect of mefenamic acid [80]. However, the evidence on stoke is highly conflicting [81].

Inflammation is a known driver for neuronal degeneration [82] and it has been shown that the incidence of Alzheimer’s disease is lower in patients with rheumatoid arthritis receiving chronic NSAID treatment [83]. However, if there is a neuroprotective effect through NSAIDs in diseases such as Alzheimer’s or Parkinson’s, this has not been conclusively established and the evidence is conflicting [69].

Adverse effects due to overuse or overdose have also been reported, e.g., medication overuse headache, ataxia, vertigo, dizziness, agitation, encephalopathy, depression, disorientation, and more [69,71,84,85]. Many of these symptoms have been reported for both mefenamic acid and ibuprofen [69,70,71,84,85,86]. It has to be noted that ibuprofen appears to be the more widely used of the two and thus the number of published reports of adverse events may appear to be higher, which can be attributed to reporting bias.

Activity of an NSAID in the brain depends on its availability in the brain. The blood–brain barrier is in the core of brain related drug pharmacokinetics. It is not surprising that the transport of NSAIDs over the blood–brain barrier differs between the different drugs [87]. Dysfunction of the blood–brain barrier due to chronic inflammation has come into the focus of neurodegeneration only recently [88].

### 5.4. Selection of the Individually Suitable NSAID

In principle, all approved NSAIDs have demonstrated their effectiveness in reducing pain and inflammation in various indications and their general tolerability is also confirmed with the existing regulatory approval. High-quality evidence in the form of RCTs, systematic reviews and meta-analyses exists mainly for osteoarthritis, rheumatoid arthritis, ankylosing spondylitis, back pain and acute gout. For the majority of indications, however, the clinical evidence is very poor, especially for high-quality and reproduced RCTs for the comparison of individual NSAIDs with each other. Figure 1 shows the imbalanced study situation of NSAIDs using the example of dysmenorrhea [89]. Almost all studies have assessed the NSAID against a placebo control, but head-to-head trials are almost non-existent. Where available, most comparative studies showed no significant differences in efficacy. Guidelines therefore do not provide recommendations for specific NSAIDs across indications and usually recommend them as a class [10].

Since NSAIDs differ mainly in their tolerability profiles and drug interactions, as explained above, the selection of an NSAID should be subject to an individual benefit–risk assessment. Factors to consider include the indication to be treated, age (older populations have a higher risk of cardiovascular events, renal dysfunction and bleeding), comorbidities (cardiovascular, kidney or gastrointestinal diseases) and the use of concomitant drugs (e.g., aspirin, anticoagulants) [10]. After weighting the respective risk, attention should be paid to the COX-1/COX-2 balance.

If the gastrointestinal risk prevails, a compound with higher COX-2 selectivity should be chosen. As an alternative pain medication, metamizole and paracetamol are suitable, especially for short-term administration, and again paracetamol or opioids for long-term administration [90,91,92]. It must be noted at this point, that opioids should only be administrated after careful consideration [9,93,94,95].

If an NSAID is required in patients with an existing risk of gastrointestinal complications, the use of a gastroprotective agent may reduce this risk (e.g., proton pump inhibitors). Ibuprofen appears to have a lower rate of gastrointestinal side effects in everyday clinical practice compared to most other NSAIDs [10,64,96,97,98]—but only at doses below 2400 mg [99]. At high cardiovascular risk, COX-2 inhibition should be rather low and naproxen is recommended. If both risks are high, COX-2-selective agents and traditional NSAIDs should be avoided altogether [10,99]. For both compounds it should be noted that no gains in analgesia can be achieved above a certain dose (ceiling effect) [15,100,101]. There is only an increased occurrence of adverse drug reactions and a generally higher risk of toxic effects. In such a case, the medication—after reaching the substance-specific maximum dose—must be switched to more potent substances.

With regard to renal and hepatic side effects, there is insufficient high-quality evidence to make a comparative assessment of the individual NSAIDs [10]. Especially for mefenamic acid, there is relatively little literature in this regard. However, it has been postulated that mefenamic acid has very complex physiological effects and in some cases achieves high intracellular concentrations, which can lead to renal and hepatic changes [16]. A certain nephrotoxicity is generally known for NSAIDs [15,56].

### 5.5. Consideration of Concomitant Medication

For elderly patients, NSAIDs are among the most frequently prescribed medications [56] and it is suspected that self-medication is prevalent [102]. Additionally, the number of concomitant medications increases with age and there thus is a risk of drug–drug interactions that can easily become complex and even unpredictable. Physicians should especially be aware of an increased bleeding risk and the potentiation of gastrointestinal adverse events when certain drug combinations are prescribed. Similar considerations may apply to chronically ill patients. Table 2 provides an overview of the interactions of NSAIDs with commonly administered drug classes. A more detailed overview of potential drug–drug interactions in elderly patients can be found in the American Geriatrics Society 2019 Updated AGS Beers Criteria^®^ [68].

### 5.6. The Special Case of Acetylsalicylic Acid

Interaction studies suggest that ibuprofen may competitively inhibit the effect of low-dose acetylsalicylic acid on platelet aggregation when both are administered simultaneously. Concomitant administration of ibuprofen and acetylsalicylic acid is generally not recommended due to the increased risk of side effects [77,78].

The binding sites of the reversible COX inhibitor ibuprofen and those of the irreversible COX inhibitor acetylsalicylic acid are adjacent to each other in the core of the COX-1 enzyme. This leads to a competitive interaction between ibuprofen and acetylsalicylic acid. When ibuprofen is taken prior to acetylsalicylic acid, it blocks access to the binding site and thus prevents the irreversible inhibition of COX-1 and thus the antithrombotic effect of acetylsalicylic acid. To avoid this problem, acetylsalicylic acid should be taken at least half an hour before or eight hours after ibuprofen. However, this intake mode is only useful for sporadic ibuprofen use [77,78]. In the case of long-term therapy, another analgesic should be used (paracetamol or COX-2 inhibitors).

Furthermore, many patients prefer more “stomach-friendly” enteric-coated acetylsalicylic acid tablets. In these patients, even a time-delayed intake is not helpful, since the time of drug delivery is too variable. Retarded acetylsalicylic acid preparations in combination with ibuprofen (or metamizole) thus do not appear to have any advantage over rapid-release formulations [77,78].

One study examined 23 NSAIDs and found that the majority of compounds interacted in vitro with the inhibitory effect of acetylsalicylic acid on thrombocyte aggregation and TXB2 formation—this was especially true of ibuprofen and mefenamic acid [103]. However, the compounds differed in their binding within the hydrophobic channel. As a common feature, all compounds that interfered with acetylsalicylic acid formed a hydrogen bond to Ser-530. Ser-530 is acetylated by acetylsalicylic acid, but the transfer of the acetyl group is prohibited by the hydrogen binding of other NSAIDs. The NSAIDs that form this hydrogen bond include ibuprofen and mefenamic acid [103].

A small observational study in patients with coronary artery disease showed that the inhibition of platelet aggregation by acetylsalicylic acid was reversed by a concomitant intake of metamizole, with the effects being reversible and dose-dependent and not occurring when acetylsalicylic acid was taken 30 min before metamizole [104]. Thus, if administered concomitantly in the short-term, metamizole should be taken 30 min after acetylsalicylic acid (at the lowest possible dose).

### 5.7. Administration during Pregnancy

In general, during the first and second trimesters, inhibitors of prostaglandin synthesis should only be taken if absolutely necessary. Care should be taken to keep the dose as low and the duration of treatment as short as possible [105].

During the third trimester of pregnancy, all prostaglandin synthesis inhibitors can expose the fetus to the risk of cardiopulmonary toxicity, with premature closure of the ductus arteriosus, pulmonary hypertension, kidney failure, intracranial hemorrhage, oligohydramnios and necrotizing enterocholitis [11,105,106]. At the end of pregnancy, even with small doses, a possible prolongation of the bleeding time may occur [11]. When used to inhibit uterine contractions in the case of premature onset of labor, these drugs can delay or prolong the birth process [11] and also lead to damage to the child [107].

For mefenamic acid, the effects on the fetus appear to have been investigated mainly on animal models. The US label speaks in this regard mainly of NSAIDs in general; mentioned mefenamic acid-specific data come exclusively from animal studies [108]. For this reason, and due to the limited data available, the Austrian label does not recommend the use of mefenamic acid in pregnant women in the first and second trimesters and is contraindicated during the third trimester of pregnancy [11].

Ibuprofen is a comparatively well-studied NSAID in this regard; however, of OTC analgesics, paracetamol appears to have the most data [109,110]. For ibuprofen, no increased risk of malformation has been documented in numerous animal studies [105,111]. The sparse evidence is mostly based on small numbers of cases, sometimes with poor investigational methodology [105,112,113,114]. One study reports that taking ibuprofen in the second trimester was significantly associated with a lower birth weight (adjusted OR 1.7, 95% CI 1.3–2.3). In addition, the use of ibuprofen in the second and third trimesters was significantly associated with asthma in 18-month-old children (adjusted OR 1.5, 95% CI 1.2–1.9; adjusted OR 1.5, 95% CI 1.1–2.1) [115]. Additionally, with ibuprofen, the rate of spontaneous abortions, as with other NSAIDs and coxibs, may be increased [116]. For dexibuprofen there is no documented evidence of its risk when used in pregnancy [105].

In summary, it can therefore be stated that ibuprofen is safe until week 28 of pregnancy [105]. The pharmacovigilance and consultation center for embryonic toxicology at Berlin’s Charité hospital (www.embryotox.de) recommends ibuprofen as the pain medication of choice alongside paracetamol during the first two trimesters. However, www.embryotox.de has no data entry on mefenamic acid.

It seems important to note again at this point: the dosage of this NSAID should always be kept as low and the duration of treatment should be as short as possible.

### 5.8. Administration during Lactation

The non-lipophilic drug ibuprofen is characterized by a short half-life of 2 h and 90–99% plasma protein binding. With a therapeutic dose of 800—1600 mg/day, the drug could not be detected in breast milk [105]. The detection limits reported in the two available studies were 1 and 0.5 mg/L, respectively [117,118].

Another study conducted several measurements of drug concentration in breast milk after administration of 400 mg of ibuprofen every 6–8 h. The drug quickly crossed into breast milk and drug concentrations of 13 ng/mL were measured just 30 min after taking the first tablet. The highest concentration of ibuprofen during the study period was determined after 20.5 h to be 181 ng/mL [119].

In addition, a comprehensive prospective study examining commonly used drugs found no side effects in breastfed children [120]. Ibuprofen therefore is considered an analgesic that can be used in lactation [106,121,122].

For mefenamic acid, unfortunately, there is only scant literature on this topic [105,106]. In an older report, it is stated that a maximum of 0.8% of relative drug dose cross into breast milk [123]. Ten newborns were breastfed for four days during maternal intake of mefenamic acid. Side effects in the infants were not reported [123,124]. There is some evidence that mefenamic acid has a longer half-life in premature infants, which is why the Committee on Drugs of the American Academy of Pediatrics advises against its use [122], but others consider its use in lactation to be justified [106].

Since there is little published evidence of mefenamic acid use during breastfeeding, other drugs should be preferred, especially during the lactation of a newborn or premature baby [124].

## 6. Summary of Considerations

This comprehensive review of the current and past literature has shown that there are hardly any comparative studies of ibuprofen versus mefenamic acid and an evidence-based evaluation of differences in the effectiveness and tolerability of these two compounds is not possible. Especially on mefenamic acid, there is hardly any recent literature. However, it is possible to take a stepwise approach in order to assess benefits and risks on an individual patient level. Table 3 provides a generic guide for such a benefit/risk assessment for NSAIDs in general with some additional comments on ibuprofen and mefenamic acid in particular, where possible.

## 7. Conclusions

It was mentioned before that most of the NSAIDs are very old drugs and that this class of compounds received renewed attention only after the discovery of underlying physiological mechanisms. At that time, mefenamic acid already was a generic drug of limited interest for industry, while ibuprofen became one of those drugs that benefited from the new mechanistic “boom”. It received a further stimulus by the upcoming “racemic drug” topic, which led to the discovery of dexibuprofen.

Comparative data of both compounds are rare. A benefit–risk assessment based solely on the degree of COX-1 or COX-2 selectivity is not permissible. Both drugs are currently approved, therefore they can be considered effective and safe in the approved dose and in the short-term setting. It is known that a ceiling effect at higher doses occurs with both drugs and no further increase in efficacy can be achieved while toxicity increases in a dose-dependent manner.

For ibuprofen there is recent literature which shows that the cardiovascular and gastrointestinal risk is low at a dosage of up to 1200 mg. In patients with a high cardiovascular risk, however, naproxen should definitely be preferred. Due to the high gastrointestinal side effect profile of naproxen, accompanying gastroprotective medication (e.g., proton pump inhibitors) must be considered. Especially in elderly patients, attention should be paid to existing risk factors and interactions with other frequently administered drugs. In the first two trimesters of pregnancy and during lactation, the safety for ibuprofen is relatively well-established and supported by recent research; for mefenamic acid, however, hardly any recent literature is available. In the third trimester, both drugs are contraindicated. Although both ibuprofen and mefenamic acid—as with the class of traditional NSAIDs in general—have been available for a very long time, conducting further comparative research to better understand the relative efficacy and tolerability of individual NSAIDs should not be neglected. Given the great importance of NSAIDs, the benefit–risk assessment should not be placed on the shoulders of the individual doctors and their individual experience and personal preference alone.

Bearing in mind that post-marketing observation of drug performance has become an indispensable element of surveillance by authorities, it may be considered a major gap in drug utilization that there is no stringent tool for the unbiased direct comparison of (sets of) drugs, not even for the most frequently used of them. Probably only a regulatory initiative could lead to an improved situation.

## Figures and Tables

**Figure 1 pharmaceutics-14-02240-f001:**
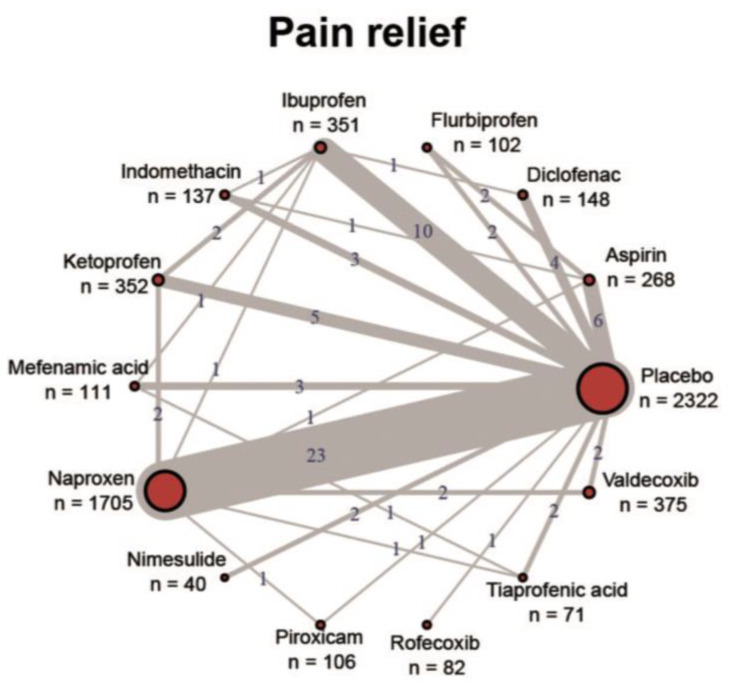
Overview of the study situation of NSAIDs using the example of dysmenorrhea. Reproduced under a Creative Commons license from [89].

**Table 1 pharmaceutics-14-02240-t001:** Pharmacokinetics of ibuprofen and mefenamic acid [10,15].

NSAIDs	Absorption	Distribution	Metabolism	Excretion	Half-Life
Mefenamic acid	BA: easily absorbed T_max_: 2–4 h	V_d_: 1.06 L/kg PB: >90%	Hepatic; mainly via CYP2C9	Renal: 52% Fecal: 20%	2–4 h
Ibuprofen	BA: 80% T_max_: 1–2 h	V_d_: 0.12-2 L/kg PB: 90–99%	Hepatic; rapidly metabolized via CYP2C9	Renal: 45–79%	1.8–2.44 h

BA, bioavailability; PB, protein binding; Tmax, time to maximum plasma concentration; Vd, volume of distribution.

**Table 2 pharmaceutics-14-02240-t002:** Interactions of NSAIDs with commonly administered drug classes.

Medication	Interaction
Antiplatelets (aspirin, clopidogrel)	Increases risk of gastrointestinal bleeding
Angiotensin-converting-enzyme inhibitor (ACEI) and Angiotensin Receptor Blockers (ARB)	Increases in blood pressure by attenuating antihypertensive effects
Beta blockers	Increases in blood pressure by attenuating antihypertensive effects
Calcium antagonists	Increases in blood pressure by attenuating antihypertensive effects
Corticosteroids	Increases risk of gastrointestinal bleeding
Digitalis glycosides	Increase serum digoxin level
Diuretics	Increases in blood pressure by attenuating antihypertensive effects
Methotrexate	NSAIDs reduce renal excretion of methotrexate, causing methotrexate toxicity.
Selective serotonin reuptake inhibitors (SSRIs)	Increases risk of gastrointestinal bleeding

Reproduced under a Creative Commons license from [56].

**Table 3 pharmaceutics-14-02240-t003:** Stepwise guide to factors to consider with NSAIDs in general, and mefenamic acid or ibuprofen in particular.

Step	Consideration	Comment
1	Indication to be treated	The approved indication for NSAIDs may differ by country, formulation and brand. There is a larger number of oral formulations and generic brands for ibuprofen than for mefenamic acid
2	Special populations	Pregnancy: only if absolutely necessary during the first and second trimester, not to be used during the third trimester; ibuprofen is well studied in pregnancy; mefenamic acid has mostly evidence from animal models Lactation: Ibuprofen is safe to be used during lactation; mefenamic acid is not well studied During pregnancy and lactation mefenamic acid is not recommended Elderly patients: the chronic, regular use of NSAIDs in elderly patients should be avoided
3	Patient age	NSAIDs are generally used in children and adults, differences in indications may apply for special populations such as newborns or preterm babies, and the elderly
4	Comorbidities	Older patients tend to have a higher risk of relevant comorbidities such as cardiovascular disease, renal disease, or gastrointestinal bleeding, which should be assessed
5	Concomitant medications	The risk of gastrointestinal bleeding is increased with antiplatelets, corticosteroids, and SSRIs. Blood pressure may be increased with ACEI/ARB, betablockers, calcium antagonists, and diuretics
6	Side effects	Gastrointestinal and cardiovascular: the risk should not be assessed based on COX-selectivity alone. A warning of gastrointestinal and cardiovascular side effects is present in the package leaflets of both ibuprofen and mefenamic acid; ibuprofen has been shown in trials to have a low gastrointestinal and cardiovascular toxicity at doses up to 1200 mg
7	Other considerations	Ibuprofen has a racemic and a pure enantiomer formulation available with different properties that may be considered in an individual patient

ACEI, angiotensin-converting-enzyme inhibitor; ARB, angiotensin receptor blocker; SSRI, selective serotonin reuptake inhibitor.

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
