# Peer review of "Interpreting the Benefit and Risk Data in Between-Drug Comparisons: Illustration of the Challenges Using the Example of Mefenamic Acid versus Ibuprofen"

_pharmaceutics, 2022, doi:10.3390/pharmaceutics14102240_

Round 1

Reviewer 1 Report

Very interesting article

Author Response

We would like to thank reviewer #1 for reviewing our article and for his/her kind feedback. No change was requested by reviewer #1.

Author Response

We would like to thank reviewer #2 for reviewing our article and for providing helpful comments. The idea of providing a risk assessment table to the readers is very important. Given the general conclusion of our article that an evidence-based recommendation on the choice between the drugs ibuprofen and mefenamic acid is not possible, we have opted for a step-by-step list of considerations that can guide clinicians when choosing an NSAID. This list has now been included as section 6, "Summary of considerations" and has a tabular format with steps 1 to 7 (Indication to be treated - Special populations - Patient age - Comorbidities - Concomitant medications - Side effects - Other considerations) and includes comments on ibuprofen and mefenamic acid and, where appropriate, NSAIDs in general.

We have also extended our section on drug-drug interactions with a focus on the elderly patient, where this issue is most relevant. We have also made reference to the most recent (2019) guidelines of the American Geriatrics Society that focusses on potentially inappropriate medications in the elderly population.

We are uploading the track-changes version of the revised article for your reference. With best wishes, André Farkouh on behalf of the authors.

Round 2

Reviewer 2 Report

The authors have addressed my comments. One minor typo correction in line 422: "and the thus" could be "and thus this". Besides that, I recommend to accept it. 

Author Response

Thank you very much for your considerate review. We have now changed "and the thus" to "and there thus", so the full sentence now reads as follows: "Additionally, the number of concomitant medications increases with age and there thus is a risk of drug-drug interactions that can easily become complex and even unpredictable."

We have also corrected another minor typo and added "for NSAIDs" to the first step in the guiding Table 3 to show that the issue of country- and formulation-specific labels is not only an issue for ibuprofen and mefenamic acid but NSAIDs in general. Especially the older drugs have been approved in Europe on a country level and not through the EMA and generic formulations often do not seek the full spectrum of indications that the original drug has.

Thank you once again for your efforts and helpful comments!

Best regards, André Farkouh
